# Silicone toothbrushes: A scoping review of an underutilized tool in global oral health

Aoife Cummins[1]*, Alexa Bennett[2], Kathryn Carrier[3], Sujay A. J. Mehta[4,5], Priyanka Gudsoorkar[6]

1 Faculty of Health Sciences, Mary Heersink's School of Global Health and Social Medicine, McMaster University, Hamilton, Ontario, Canada, 2 Department of Geography and Environmental Management, University of Waterloo, Waterloo, Ontario, Canada, 3 Faculty of Medicine, Dalhousie University, Halifax, New Brunswick, Canada, 4 Vancouver Oralfacial Pain, Vancouver, British Colombia, Canada, 5 Department of Medicine, Faculty of Health Sciences, McMaster University, Hamilton, Ontario, Canada, 6 Department of Environmental & Public Health Sciences, College of Medicine, University of Cincinnati, Cincinnati, Ohio, United States of America

☣ These authors contributed equally to this work.
* cummia2@mcmaster.ca

## Abstract

Oral diseases are the most prevalent non-communicable diseases worldwide, affecting 3.5 billion people, with a disproportionate impact on those living in low- and middle-income countries. Despite being largely preventable through proper oral hygiene, current oral health promotion strategies rely heavily on plastic and nylon toothbrushes, which present both environmental and accessibility challenges. In response to the growing need for sustainable and affordable preventive oral health solutions, there has been increasing interest in alternatives to conventional toothbrushes. This scoping review aimed to summarize the global literature on silicone toothbrushes, an underutilized tool in preventive oral care. A systematic search of five databases, supplemented by reference screening, identified ten English-language studies investigating silicone toothbrushes. Findings suggest that silicone toothbrushes are effective in plaque removal, have a lower risk of gingival trauma, are well-suited for specific populations, and perform better in environmental impact assessments. This review also demonstrated that silicone toothbrushes remain under-researched and underutilized, highlighting the need for further high-quality studies to evaluate their effectiveness, safety, and broader implementation.

## Introduction

Oral diseases are the most prevalent non-communicable diseases worldwide, affecting 3.5 billion people, with a disproportionate impact on those living in low- and middle-income countries (LMICs) [1]. These conditions often lead to adverse physical and psychosocial health impacts, including pain [2], tooth loss [3], and diminished quality of life [4]. Despite being largely preventable through proper oral hygiene,

**Data availability statement:** All data underlying the findings of this study are fully available without restriction. Table 1 provides the complete list of records identified in this scoping review, and the PRISMA diagram is shown in Fig 1. The dataset is also publicly accessible via https://doi.org/10.5281/zenodo.16900885.

**Funding:** The authors received no specific funding for this work.

**Competing interests:** The authors have declared that no competing interests exist.

current approaches to oral health promotion rely heavily on plastic and nylon toothbrushes, which pose environmental and accessibility challenges. For example, in the United States alone, approximately one billion toothbrushes are discarded annually, contributing significantly to landfill waste [5]. Moreover, for individuals with limited financial resources, the recommended practice of replacing toothbrushes every three to four months can be cost-prohibitive [1,6]. In addition, systemic inequities in access to oral health education and preventive care result in lower rates of toothbrushing among populations in LMICs [7,8].

Given the need for sustainable and affordable preventive oral health solutions, the World Dental Federation has called for the development of technologies that address environmental and health challenges [9]. Consumers and dental practitioners increasingly seeking sustainable toothbrush alternatives [10,11], including designs that minimize waste, such as replaceable heads, or use materials like recycled plastic [12], bamboo [13], and silicone [14]. Among these materials, silicone is valued for its durability, antimicrobial properties, flexibility, and ease of sterilization [15–17]. These characteristics have led to the adoption of this material in various health innovations, such as menstrual cups and medical implants [17–19]. Despite silicone's widespread use in health products, its application in oral preventive care has been limited to toothbrushes designed for children and older adults [14].

Peer-reviewed research on silicone toothbrushes is limited, generating a critical knowledge gap regarding their effectiveness, sustainability, and broader applicability in oral health. To address this gap, a comprehensive overview of existing literature is needed to evaluate the potential applications for silicone in the oral health domain. Accordingly, this paper summarizes the global literature published on silicone toothbrushes through a scoping review. Based on these findings, the potential of silicone toothbrushes as a viable solution for enhancing global oral health is explored.

## Methods

This scoping review followed the methodological framework proposed by Arksey and O'Malley (2005) and adhered to the PRISMA's Extension for Scoping Reviews (PRIMA-ScR) guidelines [20].

### Search strategy & selection of sources

A comprehensive search strategy was employed to identify peer-reviewed literature across five databases, including EMBASE, CINAHL, PUBMED, SCOPUS, and Web of Science, from inception until March 2, 2025. To ensure comprehensive coverage, Google Scholar and reference lists of relevant articles were also manually screened for additional sources. Only English-language publications were included due to resource limitations for translation. The initial search strategy was built in EMBASE and then translated to the four other electronic databases. Search terms combined controlled vocabulary and free-text keywords for sensitivity and specificity. The search syntax included variations of

"toothbrush," "oral hygiene device," "dental brush," and "silicone," with Boolean operators and proximity searching applied where possible. The following syntax was used: ("exp/ dental general device" OR "toothbrush*.mp." OR "tooth adj5 brush*.mp." OR "teeth adj5 brush*.mp." OR "oral adj5 brush*.mp." OR "dental adj5 brush*.mp." OR "tooth adj5 bristl*.mp." OR "teeth adj5 bristl*.mp." OR "oral adj5 bristl*.mp." OR "dental adj5 bristl*.mp.") AND ("silicone*.mp." OR "siloxane*.mp."). The search was limited to relevant source types (i.e., scholarly journals, books, etc.)

Articles were eligible for inclusion if they discussed a silicone-based oral health device for cleaning teeth. Articles were excluded if they focused on devices made from materials other than silicone or were used solely to polish teeth. Literature reviews were eligible if they utilized systematic approaches. The screening process was done in two-stages. Two independent reviewers conducted the abstract and full-text screening, with reviewer pairings varying across records. Studies meeting the inclusion criteria were imported into Covidence for data management. The PRISMA flowchart (Fig 1) depicts the search and selection process. Although a broad search strategy was employed, relevant studies may have been missed due to inconsistent indexing or because the material composition (e.g., silicone) was not always specified in titles or abstracts.

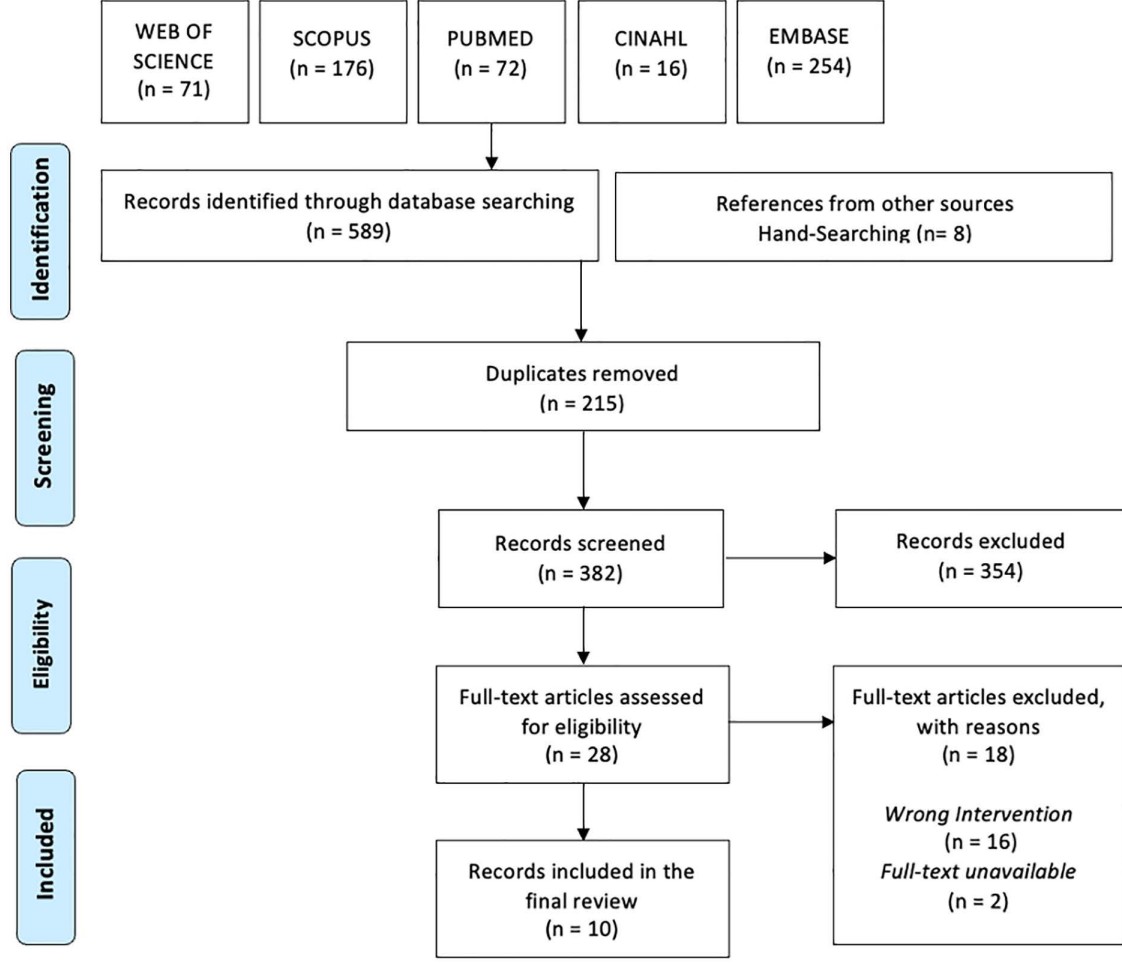

**Fig 1. PRISMA flow diagram.**

### Data extraction & synthesis

The selected articles were extracted independently using a standardized data charting form developed by the research team. The form was piloted on a subset of articles and iteratively revised to ensure consistency. Data extracted from each article included author, year of publication, study design, study population, intervention, comparison interventions, outcomes, and key conclusions. Discrepancies were resolved through discussion and, if needed, through consultation with a third reviewer to ensure consensus. A narrative synthesis summarized findings across four domains: plaque removal effectiveness, gingival health, population-specific suitability, and environmental sustainability.

These domains were selected based on the focus of the included studies and the overarching aim of assessing silicone toothbrushes as a preventive oral health tool. Data were managed using Microsoft Excel to facilitate comparison and thematic organization. Although all included studies were quantitative, they varied in methodological quality and design, including in vitro experiments, clinical trials, and life cycle assessments. No formal risk of bias assessment was conducted, given the exploratory nature of this scoping review. A summary of the findings is presented in Table 1.

## Results

### Characteristics of articles

Ten peer-reviewed articles were selected for inclusion. All articles used quantitative designs including in-vitro ($n = 2$), clinical ($n = 7$), and life-cycle assessment ($n = 1$). Of the articles with participants, the study populations included children ($n = 2$), adults ($n = 3$), and dogs ($n = 1$). Comparison groups varied and represented non-silicone toothbrushes ($n = 5$), finger brushing ($n = 1$), and a non-modified silicone swab ($n = 1$).

The articles collectively examined four dimensions of silicone toothbrushes: plaque removal effectiveness, gingival health, population-specific suitability, and environmental sustainability. These findings are illustrated in Fig 2.

### Plaque removal effectiveness

Dental plaque removal effectiveness was a central focus across most included studies [21,22,23–25,26,27]. For example, two studies found that silicone toothbrushes demonstrated comparable plaque removal to plastic and nylon toothbrushes among post-secondary students [28] and children who brushed their teeth or who had support from parents [25]. Kulkarni et al. reported that silicone finger brushes designed for babies were more effective at improving oral hygiene scores among Indian adults who typically use their fingers to clean their teeth, highlighting context-specific effectiveness [23]. Conversely, Nieri et al. found that U-shaped automatic electric toothbrushes with silicone bristles were not effective at removing plaque compared to conventional and habitual toothbrushing practices. The authors hypothesize that this finding could be due to the short length of bristles in the U-shaped device [24]. Engsomboon et al. (2024) conducted an in-vitro study to determine the optimal hardness of a newly designed silicone mouth swab for pseudo-plaque removal [21]. Their findings suggest that the optimal hardness of the silicone toothbrushes was 60 Shore A, however, there was no soft tissue damage among all hardness levels (10, 20, 30, 40, 50, and 60). In a follow-up study, Engsomboon (2025) demonstrated that increasing head length and incorporating straight, threaded bristles significantly improved the dental plaque removal of silicone toothbrushes [22]. Moreover, their findings indicate that silicone toothbrushes are equally as effective at removing plaque under both wet and dry brushing conditions.

### Gingival health

Silicone toothbrushes were associated with a reduced risk of gingival trauma and tooth abrasion across multiple studies, largely due to their soft, flexible bristle composition [21,22,25,26,27]. In an animal model, Tomofuji et al. found that, among beagle dogs, sonic toothbrushes with warmed silicone rubber bristles stimulated the highest gingival cell proliferation compared to non-warmed and nylon bristle brushes, suggesting potential regenerative or protective benefits for

**Table 1. Summary of Findings.**

| | Citation | Study Aim | Study Design | Study Setting | Population | Intervention | Comparison Intervention | Findings |
|---|---|---|---|---|---|---|---|---|
| 1 | Engsomboon et al. (2024) | To identify the ideal hardness level of silicone oral swabs for plaque removal and gingival health. | Comparative, in vitro experimental | Lab | N/A | A newly designed silicone mouth swab modified from the MouthEze | Silicone mouth swab heads with varying hardness levels: 20, 30, 40, 50, and 60 Shore A | -60 Shore A silicone oral swab optimizes pseudo-plaque removal in vitro. -Key for designing specialized oral hygiene tools for older adults, improving oral health and well-being. |
| 2 | Engsomboon et al. (2025) | To compare the pseudo-plaque removal efficiency of a new silicone mouth swab and the existing MouthEze under wet and dry conditions. | Experimental | Lab | N/A | A modified silicone mouth swab designed with straight and threaded brushing bristles | The MouthEze: silicone mouth swab with straight brushing bristles | -The new silicone swab with straight and threaded bristles and a longer head removes more pseudo-plaque than the MouthEze with only straight bristles and a shorter head. -This design may enhance oral care for individuals with limited dexterity or difficulty using traditional tools. |
| 3 | Koo et al. (2003) | To compare the plaque removal effectiveness between silicone and conventional toothbrushes. | Clinical single-use study | Brazil | Volunteer students aged 19–24 | Silicone toothbrush | Manual toothbrush with nylon bristles | -Both silicone and nylon toothbrushes effectively remove supragingival biofilms. -Silicone bristles, being softer, may reduce the risk of tooth abrasion from improper brushing techniques. |
| 4 | Kulkarni et al. (2023) | To compare the teeth cleaning effectiveness of a silicone baby finger toothbrush with finger brushing | Nonrandomized controlled trial | India | Adults aged 18–75 who clean their teeth with their finger | Baby finger toothbrush made of silicone | Finger-brushing (cleaning teeth method using finger) | -A baby finger toothbrush cleans more effectively than using fingers for oral hygiene. |
| 5 | Mazur et al. (2024) | To investigate the environmental impact and sustainability of toothbrushes made of different materials | Life Cycle Assessment | N/A | N/A | 2 manual toothbrushes with silicone bristles | 4 manual toothbrushes with nylon bristles | Silicone-bristled toothbrushes are more sustainable than nylon ones across all 18 impact categories, with an average reduction of 14%. -Improving toothbrush sustainability involves eliminating excess material, creating lighter models, and designing replaceable head options. |
| 6 | Nieri et al. (2020) | To assess the plaque removal efficacy of a U-shaped automatic toothbrush versus a powered toothbrush, habitual brushing, and no brushing | Single-use, cross-over, examiner-blind randomized controlled trial | Italy | Volunteer students aged 18–30 | U-shaped automatic electric toothbrush (UAET) with silicone bristles | Electric toothbrush with nylon bristles, habitual tooth brushing procedure, and no brushing (negative control) | UAET was ineffective in removing dental plaque, showing results as no brushing at all. -The short silicone bristles likely failed to reach the dental or gingival surfaces |
| 7 | Ozgul et al. (2019) | (1) To compare the plaque removal effectiveness of silicone tooth and gum brush with conventional toothbrush. (2) Assess the effectiveness of parent-assisted brushing versus children's self-brushing | Crossover clinical study | Not Indicated | Children aged 5–7 (with 18 partially erupted first permanent molar teeth) | Child self-brushing with silicone tooth and gum brush with double sided soft silicone bristles AND Parent-assisted brushing with silicone tooth and gum brush | Child self-brushing with manual toothbrush with nylon bristles AND Parent-assisted brushing with manual toothbrush with nylon bristles | -Both parents and children were able to remove plaque from the occlusal surface, but the type of toothbrush and who did the brushing did not make a difference. -Children under nine are generally considered to lack the motor skills needed for effective brushing, so a silicone tooth and gum brush may be recommended instead of parent-assisted brushing. |

*(Continued)*

**Table 1.** (Continued)

| | Citation | Study Aim | Study Design | Study Setting | Population | Intervention | Comparison Intervention | Findings |
|---|---|---|---|---|---|---|---|---|
| 8 | Thanal-akshme & Ramesh, (2025) | To assess the effectiveness of manual and electric toothbrushes in improving oral hygiene | Randomized controlled clinical trial | India | Blind children aged 6–12 | Standard soft-bristle manual toothbrush | MINISO electric toothbrush with soft silicone bristles | -Manual toothbrushes were more effective than electric silicone toothbrushes in improving oral health in blind children.<br><br>-The tactile feedback from manual brushing may have contributed to better gingival health and plaque removal. |
| 9 | Tomofuji et al. (2004) | To evaluate the effects of a sonic toothbrush with silicone bristles on gingival cell proliferation, compared to a manual brush and a sonic brush with nylon bristles under both standard and warmed conditions. | Experimental | Not Indicated | Beagle dogs between 1–2 years old | Manual toothbrush and sonic toothbrush with silicone rubber bristles | A manual toothbrush or an electric toothbrush with either (1) nylon bristles or (2) warmed silicone rubber | -Electric toothbrushing with warmed silicone rubber bristles resulted in maximal proliferative activity in the gingiva, thereby promoting periodontal health |
| 10 | Ustaoğlu et al. (2020) | To compare the effects of two interdental devices on plaque removal, gingival bleeding, and patient comfort and acceptance. | 2-treatment, parallel, split-mouth, examiner blind, randomized study | Turkey | Individuals aged 18–35 who were diagnosed with gingivitis | SCIP (silicone-coated interdental pick) | IDB (interdental brushes) | -Both interdental brushes and silicone coated interdental picks significantly reduce plaque removal and lower the Papillary Bleeding Index<br>-SCIP may be a better alternative to IDB, providing improved patient comfort and ease of use. |

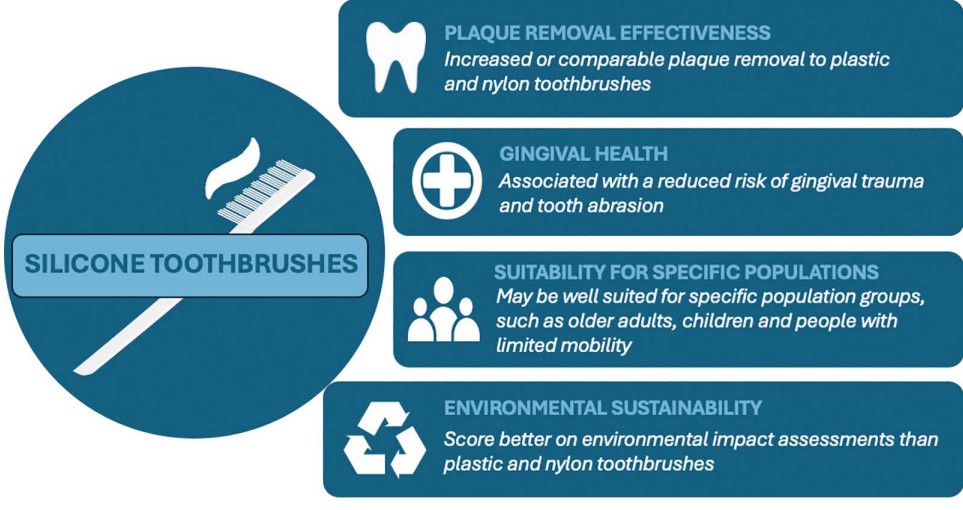

**Fig 2. Summary of findings (images sourced from: https://openclipart.org).**

the periodontal tissues [26]. Silicone toothbrushes also have the potential to benefit those with sensitive oral tissues [21]. Engsomboon et al. (2024) found that silicone toothbrushes with a hardness level of 60 Shore A effectively removed plaque without causing surface abrasions or thinning of the simulated soft tissue.

## Suitability for specific populations

Three of the studies suggest that softer silicone bristles may be well suited for specific population groups, such as older adults [21], children [25], and people with limited mobility [22]. For instance, Ozgul et al. conclude that the silicone toothbrush may be better suited for children than conventional toothbrushes as its flexible and soft design may reduce the risk of gingival traumatic injuries [25]. Thanalakshme & Ramesh demonstrated that electric silicone toothbrushes increased dental anxiety among children with blindness in comparison to soft-bristle, manual toothbrushes [29]. However, the authors suggest that this finding is attributable to fear of the unfamiliar sound of the electric motor and the challenges associated with operating the control switch. This finding reflects challenges in device interface and sensory acceptability, rather than limitations of silicone bristles. Additionally, one article included in this review also explored the effectiveness of silicone toothbrushes in both dry and wet conditions [22]. Their research showed that there was no significant difference in plaque removal whether toothbrushing was done with or without water [22]. This may indicate potential utility in settings with limited access to clean water, a critical consideration for low-resource or humanitarian contexts.

## Environmental sustainability

To investigate the environmental impact of different toothbrush materials, Mazur et al. conducted a life cycle assessment and carbon footprint analysis for 6 toothbrushes made from different materials such as nylon, polypropylene, and silicone [5]. Their results indicate that toothbrushes with silicone bristles and a polypropylene handle demonstrated superior outcomes on all 18 environmental impact categories than toothbrushes with nylon bristles and a polypropylene handle. This suggests that reducing superfluous material or replacing nylon with silicone may reduce environmental impact [5].

## Discussion

There has been little progress in the development of novel oral health strategies, with the dominant approach of plastic and nylon toothbrushes failing to address the global burden of oral diseases. This scoping review identifies that silicone toothbrushes are a promising oral hygiene tool that merits further research on both health and environmental dimensions. This review demonstrates that silicone toothbrushes are effective for plaque removal, present a lower risk of gingival injury, are suitable for specific population groups, and offer a reduced environmental impact. Based off the findings of this scoping review, this section will explore the potential health and sustainability implications of silicone toothbrushes, and considerations for its design, implementation, and application.

   This paper's findings indicate the silicone toothbrushes are to traditional, nylon toothbrushes in plaque removal, which is a key component in maintaining oral hygiene [21,22,23–25,26,27]. Oral hygiene plays a critical role in preventing conditions such as dental caries and periodontal disease, both of which are associated with pain, tooth loss, impaired function, and reduced quality of life [30]. Poor oral health also has a psychosocial impact since it restricts activities and affects self-confidence [30]. To decrease the burden of oral diseases, the WHO recommends addressing common risk factors. Dental caries, for example, stem from a diet high in free sugars, limited fluoride exposure, and inadequate mechanical plaque control [1]. In addition to other risk factors such as tobacco use, periodontal disease is closely linked to poor oral hygiene [1]. Although toothbrushing is widely endorsed as a primary prevention strategy, the global oral health community has made limited strides in developing more accessible, adaptable, or sustainable tools for maintaining oral hygiene [9,31]. Additionally, improper or inadequate toothbrushing is associated with increased risk of gingival trauma and tooth abrasion [21,22]. The results of the scoping review suggest that the soft and flexible bristle composition of silicone toothbrushes may mitigate these risks, representing a potentially more suitable oral hygiene strategy [21,22,25,26,27]. The oral health benefits of silicone toothbrushes, particularly for gingival health and plaque removal, warrant further investigation to better evaluate the effectiveness and broader applicability of silicone toothbrushes.

   An additional factor not addressed in the reviewed articles is the potential for silicone toothbrushes to exhibit enhanced antimicrobial properties compared to conventional plastic and nylon toothbrushes. Bacteria can attach to toothbrush

bristles and may be transmitted to the user, posing a risk of infection [32]. Contamination may arise from microorganisms that exist in the oral cavity, the environment, or storage containers, among other sources [32]. As silicone has antimicrobial properties, this material could decrease the risk of contamination [17]. Similarly to the silicone menstrual cup, a silicone toothbrush could be boiled for sterilization or cleaned with soap and water between uses [19]. These characteristics could make it a more suitable oral health tool for highly mobile populations or those living in poor sanitary conditions. Further, silicone materials for oral health prevention demonstrated similar effectiveness under wet and dry conditions [22] suggesting potential applicability to people with limited access to clean water and sanitation. However, further research on the diverse applications for silicone toothbrushes is needed, particularly in LMICs, to understand their potential to contribute to reducing global oral health inequities.

As highlighted by 3 papers included in this scoping review, silicone toothbrushes have shown promise in addressing the unique oral health needs of underserved populations [21,22,25]. Ozgul et al. concluded that silicone toothbrush may be better suited for children than conventional toothbrushes, as its flexible and soft design may reduce the risk of gingival traumatic injuries [25]. This aligns with their current, limited application, as silicone toothbrushes have been primarily marketed for pediatric use. While this focus emphasizes safety and comfort for children, similar benefits may extend to other populations with heightened oral health needs. For instance, Engsomboon et al. reported that silicone toothbrushes could provide gentle yet effective plaque removal for older adults [21] and individuals with reduced dexterity [22]. The benefits of silicone may also apply to people with physical disabilities or sensory sensitivities. For instance, children with autism spectrum disorder often have difficulties with oral care at home and during treatments in the dental office [33]. Sensory over-responsivity has been suggested as a factor contributing to these difficulties, particularly with toothbrushing [33]. As an added benefit, silicones' durability and reusability align with the values of environmentally conscious users, suggesting relevance across both clinical and consumer markets.

In addition to their clinical promise, the results indicate that silicone toothbrushes may offer a pathway forward in addressing environmental concerns associated with continuous plastic and nylon toothbrush disposal. Toothbrushes contribute significantly to the global burden of plastic pollution, with researchers calling for the need to consider underutilized materials to transition to a circular economy [34]. Silicone's durability can extend a product's lifespan, thereby reducing the number of toothbrushes discarded in landfills. Silicone also offers greater potential for recycling and repurposing compared to conventional plastics [34]. Medical-grade silicone has been used in health products in other sectors - such as silicone menstrual cups -which have demonstrated the lowest environmental impact scores in life cycle assessments compared to other menstrual products [35]. Silicone toothbrushes may offer similar environmental benefits; however, only one study in the present review examined the life cycle of silicone toothbrushes [5]. While the environmental impacts of sustainable toothbrushes, such as those made from bamboo and those with replaceable heads, have been increasingly studied [36], further comparative research is needed to establish silicone toothbrushes' relative advantage to traditional toothbrushes across diverse product life cycles and environmental impact settings.

Notably, no papers in this review assessed the acceptance or user perception of silicone toothbrushes. The WHO acknowledges that besides the conventional plastic and nylon toothbrush, there are diverse oral hygiene strategies used globally [37]. Indeed, oral hygiene practices are impacted by a range of factors, including cultural norms, living conditions, literacy levels, and socioeconomic status [38]. Chewing sticks are used in some regions of the world as an oral hygiene strategy [37,39]. In India, specifically within rural and semi-urban communities such as slums, people commonly rely on alternatives such as charcoal- and tobacco-based toothpowders, tree bark, or simply the use of water and a finger for cleaning [38]. Recognizing and respecting these practices is essential when developing any new oral hygiene tool. For example, a silicone finger brush may be more acceptable and culturally appropriate in communities where brushing one's teeth with a finger is currently the norm. This rationale underpinned a study by Kulkarni et al., which investigated the effectiveness of silicone finger brushes compared to the practice of finger brushing in India [23].

While this review highlights promising opportunities for silicone toothbrushes as a novel global oral health tool, it also reveals significant gaps in research on areas such as silicone toothbrush effectiveness, product design, consumer adoption, and accessibility. This scarcity of research may be indicative of a broader neglect: the undervaluation of oral health prevention strategies in global health agendas. For instance, despite clear linkages with the Sustainable Development Goals (SDGs), including SDG 3 (health and wellbeing), SDG 6 (safe water, sanitation, and hygiene), and SDG 13 (climate action), oral health remains underrepresented in research funding and policy discussions through a sustainable development lens [40]. Moreover, research and development on preventative oral hygiene receive drastically less attention than hand-hygiene or feminine hygiene in global development frameworks. For a transformation in the global oral health sector to occur, preventive oral care must be recognized as a global health priority and integrated into broader hygiene and health promotion strategies [41].

For instance, moving towards equitable global oral health means recognizing that there are no 'one-size-fits-all' solutions. There is growing recognition towards co-developing technologies in alignment with user preferences, local cultural norms, and infrastructural capacities [42,43]. Many health innovations are designed in high-income countries and introduced into LMICs with limited adaptation to local realities. This 'magic bullet' mentality is rooted in the assumption that technology alone can solve social and health disparities. This ideology can hinder effective responses to challenges in LMICs and foster narratives that blame local contexts when a product or service fails to transfer, rather than the technology itself [44–46]. Hence, to support the effectiveness of silicone toothbrushes and overall user engagement, strategies such as needs assessments, collaboration with end-users, and iterative prototyping must be adopted in the design process [42,43]. As demonstrated in several of the articles in this review, users have different oral hygiene needs and behaviors. For instance, Kulkarni et al. discussed how many individuals living in rural India use their fingers to brush their teeth, while Engsomboon et al. (2024; 2025) discussed hand dexterity issues among older adults. Designing with these realities in mind can help ensure that silicone toothbrushes are not only effective but also accessible, acceptable, and aligned with users' everyday lives [21,22,23].

Prioritizing preventive oral care also means reimagining current approaches to address oral health challenges, which are most prevalent among populations with limited resources. Currently, much of the funding and attention in global oral health is directed toward high-cost, one-time interventions, often leaving preventive care underfunded and overlooked [1,31]. In low-resource settings, a longstanding challenge has been the short lifespan of plastic and nylon toothbrushes, which can create an unsustainable dependence on external aid, supply chains, and donations. The development of longer-lasting medical-grade silicone toothbrushes offers a promising alternative. However, further research is needed to assess their real-world feasibility, affordability, and cost-effectiveness, particularly in LMICs. Studies should explore how such tools could complement treatment-based models and support long-term preventive care at the community level. In high-income contexts, preventive oral care is often embedded within routine health practices and widely available through established healthcare systems. However, access to these services is not equitably distributed, within and between countries. Populations experiencing socioeconomic disadvantages are significantly less likely to access dental care, face barriers to regularly replacing oral hygiene tools, and are disproportionately exposed to unfavorable behavioral and structural factors such as higher consumption of sugar-sweetened beverages [47,48]. These disparities contribute to a higher prevalence of oral diseases within marginalized groups and reflect broader inequities in health care access and disease outcomes. Targeted investment in development and implementation of innovative solutions is essential to make preventive oral health care more accessible, affordable, and sustainable for all.

## Limitations

Despite the use of a systematic approach, this scoping review has limitations. First, the articles were limited to peer-reviewed English articles, thus potentially missing articles written in other languages. Second, no formal quality appraisal was conducted due to the limited number and the methodological variability of the included studies. Third, four of the

included articles were identified through hand-searching, indicating that other oral health literature may not be captured in the five databases searched and silicone toothbrush-related studies may be inconsistently indexed in standard databases. Similarly, the material of interventions may not always be indicated in the abstract, title, and keywords. This may have led to omission of eligible literature that could have supported this paper's aim, for instance Kumar et al.'s paper exploring the effectiveness of a finger toothbrush [49]. Fourth, four of the included articles were either conducted in laboratory settings, involved non-human participants, or were limited to life cycle assessment, restricting the ability of this review' to evaluate the real-world effectiveness and applicability of silicone toothbrushes. Finally, the published literature on silicone toothbrushes is sparse and varies considerably in design, population characteristics, and outcome measures, which limits generalizability and the ability to draw firm conclusions. Overall, the heterogeneity and sparsity of literature highlight the need for more robust, comparative, and contextually grounded research.

## Conclusion

This scoping review mapped the global literature on silicone toothbrushes and assessed their potential as an emerging, sustainable oral hygiene tool for preventive oral healthcare. Results suggest that silicone toothbrushes are effective at plaque removal, lower risk of gingival trauma, and score better on environmental impact assessments. These attributes may make silicone toothbrushes particularly well-suited for underserved populations, individuals with specific oral health needs, and environmentally conscious consumers. This demonstrates that silicone toothbrushes remain under-researched and underutilized. Advancing their development will require targeted investment in clinical trials, design research, and implementation studies across diverse settings. As oral health gains greater recognition within the global health agenda, innovations like silicone toothbrushes offer a pathway to more equitable, sustainable, and user-centered approaches to hygiene and disease prevention.

## Supporting information

**S1 Checklist. PRISMA checklist.** From: Tricco AC, Lillie E, Zarin W, O'Brien KK, Colquhoun H, Levac D, et al. PRISMA Extension for Scoping Reviews (PRISMAScR): Checklist and Explanation. Ann Intern Med. 2018;169:467–473. https://doi.org/10.7326/M18-0850.
(PDF)

## Author contributions

**Conceptualization:** Aoife Cummins, Alexa Bennett, Kathryn Carrier, Sujay A. J. Mehta, Priyanka Gudsoorkar.

**Data curation:** Aoife Cummins, Alexa Bennett, Kathryn Carrier.

**Formal analysis:** Aoife Cummins, Alexa Bennett, Kathryn Carrier.

**Investigation:** Aoife Cummins, Alexa Bennett, Kathryn Carrier.

**Methodology:** Aoife Cummins, Alexa Bennett, Kathryn Carrier.

**Project administration:** Aoife Cummins, Alexa Bennett, Kathryn Carrier.

**Software:** Aoife Cummins, Alexa Bennett.

**Supervision:** Sujay A. J. Mehta, Priyanka Gudsoorkar.

**Validation:** Aoife Cummins, Alexa Bennett, Kathryn Carrier.

**Visualization:** Aoife Cummins, Alexa Bennett, Kathryn Carrier.

**Writing – original draft:** Aoife Cummins, Alexa Bennett, Kathryn Carrier.

**Writing – review & editing:** Aoife Cummins, Alexa Bennett, Kathryn Carrier, Priyanka Gudsoorkar.

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
