## [Decision Letter · Decision Letter 0]

16 Oct 2025

PGPH-D-25-02407

Silicone toothbrushes: A scoping review of an underutilized tool in global oral health

Dear Dr. Aoife Cummins,

Thank you for submitting your manuscript to PLOS Global Public Health. After careful consideration, we feel that it has merit but does not fully meet PLOS Global Public Health’s publication criteria as it currently stands. Therefore, we invite you to submit a revised version of the manuscript that addresses the points raised during the review process.

Your manuscript has been reviewed, and the reviewers have provided constructive comments to enhance the quality of your study. In addition to their feedback, please address the following:

Section Numbering - Remove all section and subsection numbering throughout the manuscript (e.g., 3.2, 4.1, 4.2). The paper should follow a clean format without numerical headings.

Subheadings Usage**-** Subheadings should appear Subheadings should appear Subheadings should appear Subheadings should appear **only** in the Methods and Results sections. All subheadings in the Discussion section must be removed. Instead, structure the discussion as a continuous narrative guided by your key findings.in the Methods and Results sections. All subheadings in the Discussion section must be removed. Instead, structure the discussion as a continuous narrative guided by your key findings.in the Methods and Results sections. All subheadings in the Discussion section must be removed. Instead, structure the discussion as a continuous narrative guided by your key findings.in the Methods and Results sections. All subheadings in the Discussion section must be removed. Instead, structure the discussion as a continuous narrative guided by your key findings.

Discussion Structure**-** Begin the Discussion section with a paragraph that presents the overall message of your findings. Each subsequent paragraph should: Start with a key message derived from your results, discuss similarities and differences with existing literature, and end with the public health and/or sustainability implications of the findings. Notably, the content currently under sections 4.4, 4.5 (first paragraph), and 5 is well-structured and aligns with academic standards for discussion. However, these subheadings should be removed to maintain consistency with the formatting guidelines.Begin the Discussion section with a paragraph that presents the overall message of your findings. Each subsequent paragraph should: Start with a key message derived from your results, discuss similarities and differences with existing literature, and end with the public health and/or sustainability implications of the findings. Notably, the content currently under sections 4.4, 4.5 (first paragraph), and 5 is well-structured and aligns with academic standards for discussion. However, these subheadings should be removed to maintain consistency with the formatting guidelines.Begin the Discussion section with a paragraph that presents the overall message of your findings. Each subsequent paragraph should: Start with a key message derived from your results, discuss similarities and differences with existing literature, and end with the public health and/or sustainability implications of the findings. Notably, the content currently under sections 4.4, 4.5 (first paragraph), and 5 is well-structured and aligns with academic standards for discussion. However, these subheadings should be removed to maintain consistency with the formatting guidelines.Begin the Discussion section with a paragraph that presents the overall message of your findings. Each subsequent paragraph should: Start with a key message derived from your results, discuss similarities and differences with existing literature, and end with the public health and/or sustainability implications of the findings. Notably, the content currently under sections 4.4, 4.5 (first paragraph), and 5 is well-structured and aligns with academic standards for discussion. However, these subheadings should be removed to maintain consistency with the formatting guidelines.

Limitation Section**-** Please revise the following sentence for clarity: Please revise the following sentence for clarity: Please revise the following sentence for clarity: Please revise the following sentence for clarity: *“This may have led to omission of eligible literature that could have supported this paper’s aim (i.e. (49)).”* The reference to “(i.e. (49))” is unclear and likely erroneous. Consider rephrasing or removing the parenthetical citation.The reference to “(i.e. (49))” is unclear and likely erroneous. Consider rephrasing or removing the parenthetical citation.The reference to “(i.e. (49))” is unclear and likely erroneous. Consider rephrasing or removing the parenthetical citation.The reference to “(i.e. (49))” is unclear and likely erroneous. Consider rephrasing or removing the parenthetical citation.

In the Introduction section, the authors state that the study will present findings on the effectiveness, sustainability, and accessibility of silicone toothbrushes as an alternative to traditional approaches. However, the Results section does not clearly reflect findings related to accessibility. Please clarify the following:

- Were any findings presented that compare access to silicone toothbrushes versus traditional toothbrushes?

- Did the study identify any specific groups (e.g., by age, sex, location, or socioeconomic status) with greater or lesser access to silicone toothbrushes?

- If accessibility was not directly assessed or reported, consider revising the Introduction to reflect the actual scope of the findings. Alternatively, if relevant data exist, please incorporate and highlight accessibility-related results in the Results section.

This clarification is essential to ensure consistency between the stated objectives and the reported findings.

We look forward to receiving your revised manuscript.

Kind regards,

Ifunanya Clara Agu

Academic Editor

Journal Requirements:

1. We note that this manuscript is a scoping review; our author guidelines require that you upload a PRISMA-P checklist as supporting information. Information about the PRISMA guidance and blank checklists can be found here:https://www.prisma-statement.org/protocols.

Additional Editor Comments (if provided):

Reviewers' comments:

Reviewer's Responses to Questions

**Comments to the Author**

1. Does this manuscript meet PLOS Global Public Health’s publication criteria? Is the manuscript technically sound, and do the data support the conclusions? The manuscript must describe methodologically and ethically rigorous research with conclusions that are appropriately drawn based on the data presented.? Is the manuscript technically sound, and do the data support the conclusions? The manuscript must describe methodologically and ethically rigorous research with conclusions that are appropriately drawn based on the data presented.

Reviewer #1: Yes

Reviewer #2: Yes

Reviewer #3: Yes

2. Has the statistical analysis been performed appropriately and rigorously?

Reviewer #1: Yes

Reviewer #2: No

Reviewer #3: No

3. Have the authors made all data underlying the findings in their manuscript fully available (please refer to the Data Availability Statement at the start of the manuscript PDF file)?

The PLOS Data policy requires authors to make all data underlying the findings described in their manuscript fully available without restriction, with rare exception. The data should be provided as part of the manuscript or its supporting information, or deposited to a public repository. For example, in addition to summary statistics, the data points behind means, medians and variance measures should be available. If there are restrictions on publicly sharing data—e.g. participant privacy or use of data from a third party—those must be specified.requires authors to make all data underlying the findings described in their manuscript fully available without restriction, with rare exception. The data should be provided as part of the manuscript or its supporting information, or deposited to a public repository. For example, in addition to summary statistics, the data points behind means, medians and variance measures should be available. If there are restrictions on publicly sharing data—e.g. participant privacy or use of data from a third party—those must be specified.

Reviewer #1: Yes

Reviewer #2: No

Reviewer #3: Yes

4. Is the manuscript presented in an intelligible fashion and written in standard English?

Reviewer #1: Yes

Reviewer #2: Yes

Reviewer #3: Yes

Reviewer #1: Dear Authors,

Your work is excellent and well written. The topic is really timely and very interesting. The environmental impact of Dentistry will be the topic of our near future. My recommendation is that it be accepted in its present form.

Kind regards.

Reviewer #2: Is Google Scholar an established database for grey literature?

the search strategy used was for which database?

Which authors did the data extraction and who solved the discrepancies?

Sample size and outcome variables needed in the data extraction table.

Overall number of participants in each comparison wherever applicable is needed.

Possibility of meta-analysis can be assessed if outcome data is available.

Image credits for figure 2 needed.

Overall, due to aforementioned short comings, the review can be considered incomplete and evidence may not be strong enough for a recommendation.

Reviewer #3: Overall, this manuscript addresses an interesting and timely topic on the potential role of silicone toothbrushes in promoting sustainable oral health care. The paper is generally well-structured and written, and the topic has relevance for both clinical practice and global public health. However, several areas of the manuscript could be strengthened to improve clarity, coherence, and methodological transparency.

1. While the introduction effectively highlights the global burden of oral diseases and the environmental concerns related to conventional toothbrushes, the emphasis on low- and middle-income countries (LMICs) as being disproportionately affected may need reconsideration. Oral diseases are indeed a global health issue, with high prevalence and significant impact across all income levels. The framing around LMICs might inadvertently narrow the perceived scope of the problem.

2. The second paragraph of the Introduction currently reads as a list of materials rather than a cohesive narrative linking sustainability concerns to the specific potential of silicone. I recommend that the authors more clearly articulate why silicone, compared to other sustainable materials, may offer unique advantages for oral health applications. Additionally, the final sentences could better highlight the specific knowledge gap and research need that justify this scoping review.

3. Although a formal risk of bias assessment is not mandatory for scoping reviews, it is good practice to include a brief reflection on the methodological variability or potential limitations of the included studies. This would help readers interpret the strength and consistency of the available evidence.

**Do you want your identity to be public for this peer review?** For information about this choice, including consent withdrawal, please see our Privacy Policy..

Reviewer #1: No

Reviewer #2: No

Reviewer #3: No

---

## [Decision Letter · Decision Letter 1]

19 Feb 2026

Silicone toothbrushes: A scoping review of an underutilized tool in global oral health

PGPH-D-25-02407R1

Dear Ms. Cummins,

We are pleased to inform you that your manuscript 'Silicone toothbrushes: A scoping review of an underutilized tool in global oral health' has been provisionally accepted for publication in PLOS Global Public Health.

Best regards,

Julia Robinson

Executive Editor

Reviewer Comments (if any, and for reference):

Reviewer's Responses to Questions

**Comments to the Author**

Reviewer #1: All comments have been addressed

publication criteria? Is the manuscript technically sound, and do the data support the conclusions? The manuscript must describe methodologically and ethically rigorous research with conclusions that are appropriately drawn based on the data presented.? Is the manuscript technically sound, and do the data support the conclusions? The manuscript must describe methodologically and ethically rigorous research with conclusions that are appropriately drawn based on the data presented.

Reviewer #1: Yes

3. Has the statistical analysis been performed appropriately and rigorously?

Reviewer #1: N/A

4. Have the authors made all data underlying the findings in their manuscript fully available (please refer to the Data Availability Statement at the start of the manuscript PDF file)?

The PLOS Data policy requires authors to make all data underlying the findings described in their manuscript fully available without restriction, with rare exception. The data should be provided as part of the manuscript or its supporting information, or deposited to a public repository. For example, in addition to summary statistics, the data points behind means, medians and variance measures should be available. If there are restrictions on publicly sharing data—e.g. participant privacy or use of data from a third party—those must be specified.requires authors to make all data underlying the findings described in their manuscript fully available without restriction, with rare exception. The data should be provided as part of the manuscript or its supporting information, or deposited to a public repository. For example, in addition to summary statistics, the data points behind means, medians and variance measures should be available. If there are restrictions on publicly sharing data—e.g. participant privacy or use of data from a third party—those must be specified.

Reviewer #1: Yes

5. Is the manuscript presented in an intelligible fashion and written in standard English?

Reviewer #1: Yes

Reviewer #1: Dear Authors,

thank you for the revisions and for improving the overall structure and readability of your manuscript. The text is now clearer, more coherent, and easier to follow. I believe that the changes you have implemented have strengthened the presentation of your work and enhanced its scientific value.

In light of these improvements, I am pleased to recommend acceptance of your manuscript in its current form for publication.

Best regards.

**Do you want your identity to be public for this peer review?** For information about this choice, including consent withdrawal, please see our Privacy Policy..

Reviewer #1: No
